# Ensemble Distillation for Robust Model Fusion in Federated Learning

**Tao Lin**[*]**, Lingjing Kong**[*]**, Sebastian U. Stich, Martin Jaggi.**
MLO, EPFL, Switzerland
{tao.lin, lingjing.kong, sebastian.stich, martin.jaggi}@epfl.ch

## Abstract

Federated Learning (FL) is a machine learning setting where many devices collaboratively train a machine learning model while keeping the training data decentralized. In most of the current training schemes the central model is refined by averaging the parameters of the server model and the updated parameters from the client side. However, directly averaging model parameters is only possible if all models have the same structure and size, which could be a restrictive constraint in many scenarios.

In this work we investigate more powerful and more flexible aggregation schemes for FL. Specifically, we propose ensemble distillation for model fusion, i.e. training the central classifier through unlabeled data on the outputs of the models from the clients. This knowledge distillation technique mitigates privacy risk and cost to the same extent as the baseline FL algorithms, but allows flexible aggregation over heterogeneous client models that can differ e.g. in size, numerical precision or structure. We show in extensive empirical experiments on various CV/NLP datasets (CIFAR-10/100, ImageNet, AG News, SST2) and settings (heterogeneous models/data) that the server model can be trained much faster, requiring fewer communication rounds than any existing FL technique so far.

## 1 Introduction

Federated Learning (FL) has emerged as an important machine learning paradigm in which a federation of clients participate in collaborative training of a centralized model [62, 51, 65, 8, 5, 42, 34]. The clients send their model parameters to the server but never their private training datasets, thereby ensuring a basic level of privacy. Among the key challenges in federated training are communication overheads and delays (one would like to train the central model with as few communication rounds as possible), and client heterogeneity: the training data (non-i.i.d.-ness), as well as hardware and computing resources, can change drastically among clients, for instance when training on commodity mobile devices.

Classic training algorithms in FL, such as federated averaging (FEDAVG) [51] and its recent adaptations [53, 44, 25, 35, 26, 58], are all based on directly averaging of the participating client's *parameters* and can hence only be applied if all client's models have the same size and structure. In contrast, ensemble learning methods [77, 15, 2, 14, 56, 47, 75] allow to combine multiple heterogeneous weak classifiers by averaging the *predictions* of the individual models instead. However, applying ensemble learning techniques directly in FL is infeasible in practice due to the large number of participating clients, as it requires keeping weights of all received models on the server and performing naive ensembling (logits averaging) for inference.

To enable federated learning in more realistic settings, we propose to use ensemble distillation [7, 22] for robust model fusion (FedDF). Our scheme leverages unlabeled data or artificially generated examples (e.g. by a GAN's generator [17]) to aggregate knowledge from all received (heterogeneous)

---

[*]Equal contribution.

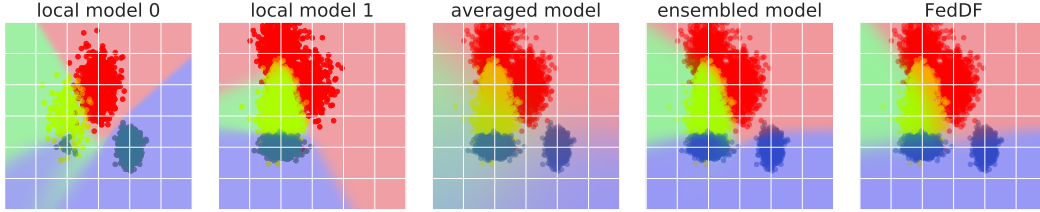

| local model 0 | local model 1 | averaged model | ensembled model | FedDF |

Figure 1: **Limitations of FEDAVG.** We consider a toy example of a 3-class classification task with a 3-layer MLP, and display the decision boundaries (probabilities over RGB channels) on the input space. The left two figures show the individually trained local models. The right three figures evaluate aggregated models and the global data distribution; the averaged model results in much blurred decision boundaries. The used datasets are displayed in Figure 8 (Appendix C.1).

client models. We demonstrate with thorough empirical results that our ensemble distillation approach not only addresses the existing quality loss issue [24] of Batch Normalization (BN) [31] for networks in a homogeneous FL system, but can also break the knowledge barriers among heterogeneous client models. Our main contributions are:

- We propose a distillation framework for robust federated model fusion, which allows for heterogeneous client models and data, and is robust to the choices of neural architectures.
- We show in extensive numerical experiments on various CV/NLP datasets (CIFAR-10/100, ImageNet, AG News, SST2) and settings (heterogeneous models and/or data) that the server model can be trained much faster, requiring fewer communication rounds than any existing FL technique.

We further provide insights on when FedDF can outperform FEDAVG (see also Fig. 1 that highlights an intrinsic limitation of parameter averaging based approaches) and what factors influence FedDF.

## 2 Related Work

**Federated learning.** The classic algorithm in FL, FEDAVG [51], or local SGD [46] when all devices are participating, performs weighted parameter average over the client models after several local SGD updates with weights proportional to the size of each client's local data. Weighting schemes based on client loss are investigated in [53, 44]. To address the difficulty of directly averaging model parameters, [64, 74] propose to use optimal transport and other alignment schemes to first align or match individual neurons of the neural nets layer-wise before averaging the parameters. However, these layer-based alignment schemes necessitate client models with the same number of layers and structure, which is restrictive in heterogeneous systems in practice.

Another line of work aims to improve local client training, i.e., client-drift problem caused by the heterogeneity of local data [43, 35]. For example, FEDPROX [43] incorporates a proximal term for the local training. Other techniques like acceleration, recently appear in [25, 26, 58].

**Knowledge distillation.** Knowledge distillation for neural networks is first introduced in [7, 22]. By encouraging the student model to approximate the output logits of the teacher model, the student is able to imitate the teacher's behavior with marginal quality loss [59, 79, 36, 71, 37, 28, 1, 70]. Some work study the ensemble distillation, i.e., distilling the knowledge of an ensemble of teacher models to a student model. To this end, existing approaches either average the logits from the ensemble of teacher models [77, 15, 2, 14], or extract knowledge from the feature level [56, 47, 75].

Most of these schemes rely on using the original training data for the distillation process. In cases where real data is unavailable, some recent work [54, 52] demonstrate that distillation can be accomplished by crafting pseudo data either from the weights of the teacher model or through a generator adversarially trained with the student. FedDF can be combined with all of these approaches. In this work, we consider unlabeled datasets for ensemble distillation, which could be either collected from other domains or directly generated from a pre-trained generator.

**Comparison with close FL work.** Guha *et al*. [18] propose "one-shot fusion" through unlabeled data for SVM loss objective, whereas we consider multiple-round scenarios on diverse neural architectures and tasks. FD [33] utilizes distillation to reduce FL communication costs. To this end, FD synchronizes logits per label which are accumulated during the local training. The averaged logits per label (over local steps and clients) will then be used as a distillation regularizer for the next round's local training. Compared to FEDAVG, FD experiences roughly 15% quality drop on MNIST. In contrast, FedDF shows superior learning performance over FEDAVG and can significantly reduce the number of communication rounds to reach target accuracy on diverse challenging tasks.

FedMD [41] and the recently proposed Cronus [9] consider learning through averaged logits per sample on a public dataset. After the initial pre-training on the labeled public dataset, FedMD learns on the public and private dataset iteratively for personalization, whereas in Cronus, the public dataset (with soft labels) is used jointly with local private data for the local training. As FedMD trains client models simultaneously on both labeled public and private datasets, the model classifiers have to include all classes from both datasets. Cronus, in its collaborative training phase, mixes public and private data for local training. Thus for these methods, the public dataset construction requires careful deliberation and even prior knowledge on clients' private data. Moreover, how these modifications impact local training quality remains unclear. FedDF faces no such issues: we show that FedDF is robust to distillation dataset selection and the distillation is performed on the server side, leaving local training unaffected. We include a detailed discussion with FedMD, Cronus in Appendix A. When preparing this version, we also notice other contemporary work [68, 10, 81, 19] and we defer discussions to Appendix A.

## 3 Ensemble Distillation for Robust Model Fusion

---

**Algorithm 1** Illustration of FedDF on $K$ homogeneous clients (indexed by $k$) for $T$ rounds, $n_k$ denotes the number of data points per client and $C$ the fraction of clients participating in each round. The server model is initialized as $\mathbf{x}_0$. While FEDAVG just uses the averaged models $\mathbf{x}_{t,0}$, we perform $N$ iterations of server-side model fusion on top (line 7 – line 10).

---

1: **procedure** SERVER
2:     **for** each communication round $t = 1, \ldots, T$ **do**
3:         $S_t \leftarrow$ random subset ($C$ fraction) of the $K$ clients
4:         **for** each client $k \in S_t$ **in parallel do**
5:             $\hat{\mathbf{x}}_t^k \leftarrow$ Client-LocalUpdate$(k, \mathbf{x}_{t-1})$         ▷ detailed in Algorithm 2.
6:         initialize for model fusion $\mathbf{x}_{t,0} \leftarrow \sum_{k \in S_t} \frac{n_k}{\sum_{k \in S_t} n_k} \hat{\mathbf{x}}_t^k$
7:         **for** $j$ in $\{1, \ldots, N\}$ **do**
8:             sample a mini-batch of samples $\mathbf{d}$, from e.g. (1) an unlabeled dataset, (2) a generator
9:             use ensemble of $\{\hat{\mathbf{x}}_t^k\}_{k \in S_t}$ to update server student $\mathbf{x}_{t,j-1}$ through AVGLOGITS
10:         $\mathbf{x}_t \leftarrow \mathbf{x}_{t,N}$
11:     **return** $\mathbf{x}_T$

---

In this section, we first introduce the core idea of the proposed Federated Distillation Fusion (FedDF). We then comment on its favorable characteristics and discuss possible extensions.

**Ensemble distillation.** We first discuss the key features of FedDF for the special case of homogeneous models, i.e. when all clients share the same network architecture (Algorithm 1). For model fusion, the server distills the ensemble of $|S_t|$ client teacher models to one single server student model. For the distillation, the teacher models are evaluated on mini-batches of unlabeled data on the server (forward pass) and their logit outputs (denoted by $f(\hat{\mathbf{x}}_t^k, \mathbf{d})$ for mini-batch $\mathbf{d}$) are used to train the student model on the server:

$$\mathbf{x}_{t,j} := \mathbf{x}_{t,j-1} - \eta \frac{\partial \mathrm{KL}\left(\sigma\left(\frac{1}{|S_t|} \sum_{k \in S_t} f(\hat{\mathbf{x}}_t^k, \mathbf{d})\right), \sigma\left(f(\mathbf{x}_{t,j-1}, \mathbf{d})\right)\right)}{\partial \mathbf{x}_{t,j-1}} . \qquad \text{(AVGLOGITS)}$$

Here KL stands for Kullback–Leibler divergence, $\sigma$ is the softmax function, and $\eta$ is the stepsize. FedDF can easily be extended to heterogeneous FL systems (Algorithm 3 and Figure 7 in Appendix B). We assume the system contains $p$ distinct model prototype groups that potentially differ in neural architecture, structure and numerical precision. By ensemble distillation, each model architecture group acquires knowledge from logits averaged over *all* received models, thus mutual beneficial information can be shared across architectures; in the next round, each activated client receives the corresponding fused prototype model. Notably, as the fusion takes place on the server side, there is no additional burden and interference on clients.

**Utilizing unlabeled/generated data for distillation.** Unlike most existing ensemble distillation methods that rely on *labeled* data from the training domain, we demonstrate the feasibility of achieving model fusion by using *unlabeled* datasets from other domains for the sake of privacy-preserving FL. Our proposed method also allows the use of synthetic data from a pre-trained generator (e.g.

GAN[2]) as distillation data to alleviate potential limitations (e.g. acquisition, storage) of real unlabeled datasets.

**Discussions on privacy-preserving extension.** Our proposed model fusion framework in its simplest form—like most existing FL methods—requires to exchange models between the server and each client, resulting in potential privacy leakage due to e.g. memorization present in the models. Several existing protection mechanisms can be added to our framework to protect clients from adversaries. These include adding differential privacy [16] for client models, or performing hierarchical and decentralized model fusion through synchronizing locally inferred logits e.g. on *random* public data[3], as in the recent work [9]. We leave further explorations of this aspect for future work.

## 4 Experiments

### 4.1 Setup

**Datasets and models.** We evaluate the learning of different SOTA FL methods on both CV and NLP tasks, on architectures of ResNet [20], VGG [63], ShuffleNetV2 [48] and DistilBERT [60]. We consider federated learning CIFAR-10/100 [38] and ImageNet [39] (down-sampled to image resolution 32 for computational feasibility [11]) from scratch for CV tasks; while for NLP tasks, we perform federated fine-tuning on a 4-class news classification dataset (AG News [80]) and a 2-class classification task (Stanford Sentiment Treebank, SST2 [66]). The validation dataset is created for CIFAR-10/100, ImageNet, and SST2, by holding out $10\%$, $1\%$ and $1\%$ of the original training samples respectively; the remaining training samples are used as the training dataset (before partitioning client data) and the whole procedure is controlled by random seeds. We use validation/test datasets on the server and report the test accuracy over three different random seeds.

**Heterogeneous distribution of client data.** We use the Dirichlet distribution as in [78, 25] to create disjoint non-i.i.d. client training data. The value of $\alpha$ controls the degree of non-i.i.d.-ness: $\alpha = 100$ mimics identical local data distributions, and the smaller $\alpha$ is, the more likely the clients hold examples from only one class (randomly chosen). Figure 2 visualizes how samples are distributed among 20 clients for CIFAR-10 on different $\alpha$ values; more visualizations are shown in Appendix C.2.

**Baselines.** FedDF is designed for effective model fusion on the server, considering the accuracy of the global model on the test dataset. Thus we omit the comparisons to methods designed for personalization (e.g. FedMD [41]), security/robustness (e.g. Cronus [9]), and communication efficiency (e.g. [33], known for poorer performance than FEDAVG). We compare FedDF with SOTA FL methods, including 1) FEDAVG [51], 2) FEDPROX [43] (for better local training under heterogeneous systems), 3) accelerated FEDAVG a.k.a. FEDAVGM[4] [25, 26], and 4) FEDMA[5] [74] (for better model fusion). We elaborate on the reasons for omitted numerical comparisons in Appendix A.

**The local training procedure.** The FL algorithm randomly samples a fraction ($C$) of clients per communication round for local training. For the sake of simplicity, the local training in our experiments uses a constant learning rate (no decay), no Nesterov momentum acceleration, and no weight decay. The hyperparameter tuning procedure is deferred to Appendix C.2. Unless mentioned otherwise the learning rate is set to $0.1$ for ResNet-like nets, $0.05$ for VGG, and $1e\!-\!5$ for DistilBERT.

**The model fusion procedure.** We evaluate the performance of FedDF by utilizing either randomly sampled data from existing (unlabeled) datasets[6] or BigGAN's generator [6]. Unless mentioned otherwise we use CIFAR-100 and downsampled ImageNet (image size 32) as the distillation datasets for FedDF on CIFAR-10 and CIFAR-100 respectively. Adam with learning rate $1e\!-\!3$ (w/ cosine annealing) is used to distill knowledge from the ensemble of received local models. We employ early-stopping to stop distillation after the validation performance plateaus for $1e3$ steps (total $1e4$ update steps). The hyperparameter used for model fusion is kept constant over all tasks.

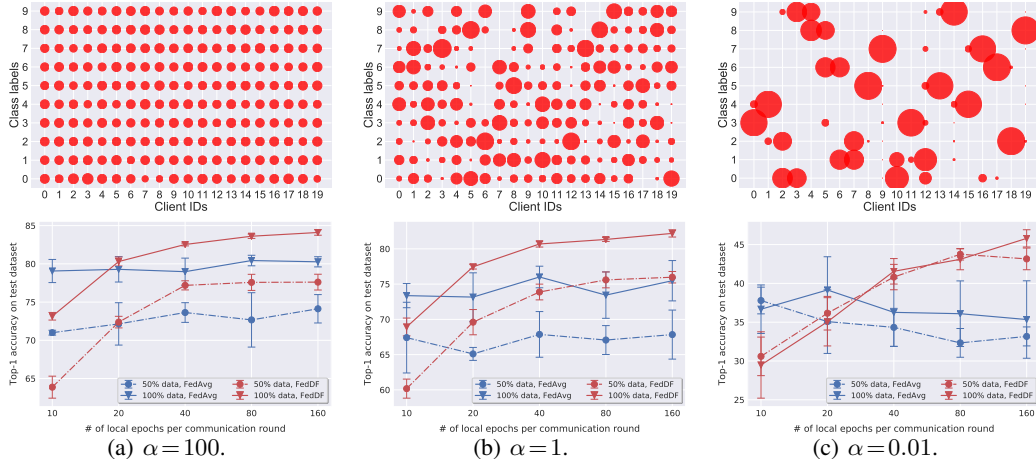

Figure 2: **Top: Illustration of # of samples per class allocated to each client** (indicated by dot sizes), for different Dirichlet distribution $\alpha$ values. **Bottom: Test performance** of **FedDF** and **FEDAVG** on **CIFAR-10** with **ResNet-8**, for different local training settings: non-i.i.d. degrees $\alpha$, data fractions, and # of local epochs per communication round. We perform 100 communication rounds, and active clients are sampled with ratio $C = 0.4$ from a total of 20 clients. Detailed learning curves in these scenarios can be found in Appendix C.4.

## 4.2 Evaluation on the Common Federated Learning Settings

**Performance overview for different FL scenarios.** We can observe from Figure 2 that FedDF consistently outperforms FEDAVG for all client fractions and non-i.i.d. degrees when the local training is reasonably sufficient (e.g. over 40 epochs).

FedDF benefits from larger numbers of local training epochs. This is because the performance of the model ensemble is highly dependent on the diversity among its individual models [40, 67]. Thus longer local training leads to greater diversity and quality of the ensemble and hence a better distillation result for the fused model. This characteristic is desirable in practice as it helps reduce the communication overhead in FL systems. In contrast, the performance of FEDAVG saturates and even degrades with the increased number of local epochs, which is consistent with observations in [51, 8, 74]. As FedDF focuses on better model fusion on the server side, it is orthogonal to recent techniques (e.g. [61, 35, 12]) targeting the issue of non-i.i.d. local data. We believe combining FedDF with these techniques can lead to a more robust FL, which we leave as future work[7].

**Ablation study of FedDF.** We provide detailed ablation study for FedDF in Appendix C.4.1 to identify the source of the benefits. For example, Table 5 justifies the importance of using the uniformly averaged local models as a starting model (line 6 in Algorithm 1 and line 11 in Algorithm 3), for the quality of ensemble distillation in FedDF. We further investigate the effect of different optimizers (for on-server ensemble distillation) on the federated learning performance in Table 6 and Table 7.

**Detailed comparison of FedDF with other SOTA federated learning methods for CV tasks.** Table 1 summarizes the results for various degrees of non-i.i.d. data, local training epochs and client sampling fractions. In all scenarios, FedDF requires significantly fewer communication rounds than other SOTA methods to reach designated target accuracies. The benefits of FedDF can be further pronounced by taking more local training epochs as illustrated in Figure 2.

All competing methods have strong difficulties with increasing data heterogeneity (non-i.i.d. data, i.e. smaller $\alpha$), while FedDF shows significantly improved robustness to data heterogeneity. In most scenarios in Table 1, the reduction of $\alpha$ from 1 to 0.1 almost triples the number of communication rounds for FEDAVG, FEDPROX and FEDAVGM to reach target accuracies, whereas less than twice the number of rounds are sufficient for FedDF.

Increasing the sampling ratio makes a more noticeable positive impact on FedDF compared to other methods. We attribute this to the fact that an ensemble tends to improve in robustness and quality, with a larger number of reasonable good participants, and hence results in better model fusion. Nevertheless, even in cases with a very low sampling fraction (i.e. $C = 0.2$), FedDF still maintains a considerable leading margin over the closest competitor.

Table 1: **Evaluating different FL methods in different scenarios** (i.e. different client sampling fractions, # of local epochs and target accuracies), in terms of **the number of communication rounds to reach target top-1 test accuracy**. We evaluate on ResNet-8 with CIFAR-10. For each communication round, a fraction $C$ of the total 20 clients are randomly selected. $T$ denotes the specified target top-1 test accuracy. Hyperparameters are fine-tuned for each method (FEDAVG, FEDPROX, and FEDAVGM); FedDF uses the optimal learning rate from FEDAVG. The performance upper bound of (tuned) centralized training is $86\%$ (trained on all local data).

| | Local epochs | The number of communication rounds to reach target performance $T$ | | | | | |
| --- | --- | --- | --- | --- | --- | --- | --- |
| | | $C=0.2$ | | $C=0.4$ | | $C=0.8$ | |
| | | $\alpha=1, T=80\%$ | $\alpha=0.1, T=75\%$ | $\alpha=1, T=80\%$ | $\alpha=0.1, T=75\%$ | $\alpha=1, T=80\%$ | $\alpha=0.1, T=75\%$ |
| FEDAVG | 1 | $350 \pm 31$ | $546 \pm 191$ | $246 \pm 41$ | $445 \pm 8$ | $278 \pm 83$ | $361 \pm 111$ |
| | 20 | $144 \pm 51$ | $423 \pm 105$ | $97 \pm 29$ | $309 \pm 88$ | $103 \pm 26$ | $379 \pm 151$ |
| | 40 | $130 \pm 13$ | $312 \pm 87$ | $104 \pm 52$ | $325 \pm 82$ | $100 \pm 76$ | $312 \pm 110$ |
| FEDPROX | 20 | $99 \pm 61$ | $346 \pm 12$ | $91 \pm 40$ | $235 \pm 41$ | $92 \pm 21$ | $237 \pm 93$ |
| | 40 | $115 \pm 17$ | $270 \pm 96$ | $87 \pm 49$ | $229 \pm 79$ | $80 \pm 44$ | $284 \pm 130$ |
| FEDAVGM | 20 | $92 \pm 15$ | $299 \pm 85$ | $92 \pm 46$ | $221 \pm 29$ | $97 \pm 37$ | $235 \pm 129$ |
| | 40 | $135 \pm 52$ | $322 \pm 99$ | $78 \pm 28$ | $224 \pm 38$ | $83 \pm 34$ | $232 \pm 11$ |
| **FedDF** (ours) | 20 | $\mathbf{61} \pm 24$ | $\mathbf{102} \pm 42$ | $\mathbf{28} \pm 10$ | $\mathbf{51} \pm 4$ | $\mathbf{22} \pm 1$ | $\mathbf{33} \pm 18$ |
| | 40 | $\mathbf{28} \pm 6$ | $\mathbf{80} \pm 25$ | $\mathbf{20} \pm 4$ | $\mathbf{39} \pm 10$ | $\mathbf{14} \pm 2$ | $\mathbf{20} \pm 4$ |

Table 2: **The impact of normalization techniques** (i.e. BN, GN) for ResNet-8 on CIFAR (20 clients with $C=0.4$, 100 communication rounds, and 40 local epochs per round). We use a constant learning rate and tune other hyperparameters. The distillation dataset of FedDF for CIFAR-100 is ImageNet (with image size of 32).

| Datasets | | Top-1 test accuracy of different methods | | | | |
| --- | --- | --- | --- | --- | --- | --- |
| | | FEDAVG, w/ BN | FEDAVG, w/ GN | FEDPROX, w/ GN | FEDAVGM, w/ GN | **FedDF**, w/ BN |
| CIFAR-10 | $\alpha=1$ | $76.01 \pm 1.53$ | $78.57 \pm 0.22$ | $76.32 \pm 1.98$ | $77.79 \pm 1.22$ | $\mathbf{80.69} \pm 0.43$ |
| | $\alpha=0.1$ | $62.22 \pm 3.88$ | $68.37 \pm 0.50$ | $68.65 \pm 0.77$ | $68.63 \pm 0.79$ | $\mathbf{71.36} \pm 1.07$ |
| CIFAR-100 | $\alpha=1$ | $35.56 \pm 1.99$ | $42.54 \pm 0.51$ | $42.94 \pm 1.23$ | $42.83 \pm 0.36$ | $\mathbf{47.43} \pm 0.45$ |
| | $\alpha=0.1$ | $29.14 \pm 1.91$ | $36.72 \pm 1.50$ | $35.74 \pm 1.00$ | $36.29 \pm 1.98$ | $\mathbf{39.33} \pm 0.03$ |

Table 3: **Top-1 test accuracy of federated learning CIFAR-10 on VGG-9 (w/o BN)**, for 20 clients with $C=0.4$, $\alpha=1$ and 100 communication rounds (40 epochs per round). We by default drop dummy predictors.

| Methods | Top-1 test accuracy @ communication round | | | | |
| --- | --- | --- | --- | --- | --- |
| | 5 | 10 | 20 | 50 | 100 |
| FEDAVG (w/o drop-worst) | $45.72 \pm 30.95$ | $51.06 \pm 35.56$ | $53.22 \pm 37.43$ | $29.60 \pm 40.66$ | $7.52 \pm 4.29$ |
| FEDMA (w/o drop-worst) [1] | $23.41 \pm 0.00$ | $27.55 \pm 0.10$ | $41.56 \pm 0.08$ | $60.35 \pm 0.03$ | $65.0 \pm 0.02$ |
| FEDAVG | $64.77 \pm 1.24$ | $70.28 \pm 1.02$ | $75.80 \pm 1.36$ | $77.98 \pm 1.81$ | $78.34 \pm 1.42$ |
| FEDPROX | $63.86 \pm 1.55$ | $71.85 \pm 0.75$ | $75.57 \pm 1.16$ | $77.85 \pm 1.96$ | $78.60 \pm 1.91$ |
| **FedDF** | $\mathbf{66.08} \pm 4.14$ | $\mathbf{72.80} \pm 1.59$ | $\mathbf{75.82} \pm 2.09$ | $\mathbf{79.05} \pm 0.54$ | $\mathbf{80.36} \pm 0.63$ |

[1] FEDMA does not support drop-worst operation due to its layer-wise communication/fusion scheme. The number of local training epochs per layer is 5 (45 epochs per model) thus results in stabilized training. More details can be found in Appendix C.2.

**Comments on Batch Normalization.** Batch Normalization (BN) [31] is the current workhorse in convolutional deep learning tasks and has been employed by default in most SOTA CNNs [20, 27, 48, 69]. However, it often fails on heterogeneous training data. Hsieh *et al.* [24] recently examined the non-i.i.d. data 'quagmire' for distributed learning and point out that replacing BN by Group Normalization (GN) [76] can alleviate some of the quality loss brought by BN due to the discrepancies between local data distributions.

As shown in Table 2, despite additional effort on architecture modification and hyperparameter tuning (i.e. the number of groups in GN), baseline methods with GN replacement still lag much behind FedDF. FedDF provides better model fusion which is robust to non-i.i.d. data, and is compatible with BN, thus avoids extra efforts for modifying the standard SOTA neural architectures. Figure 13 in Appendix C.3 shows the complete learning curves.

We additionally evaluate architectures originally designed without BN (i.e. VGG), to demonstrate the broad applicability of FedDF. Due to the lack of normalization layers, VGG is vulnerable to non-i.i.d. local distributions. We observe that received models on the server might output random prediction results on the validation/test dataset and hence give rise to uninformative results overwhelmed by large variance (as shown in Table 3). We address this issue by a simple treatment[8], "drop-worst", i.e., dropping learners with random predictions on the server validation dataset (e.g. $10\%$ accuracy for CIFAR-10), in each round before applying model averaging and/or ensemble distillation. Table 3 examines the FL methods (FEDAVG, FEDPROX, FEDMA and FedDF) on VGG-9; FedDF consistently outperforms other methods by a large margin for different communication rounds.

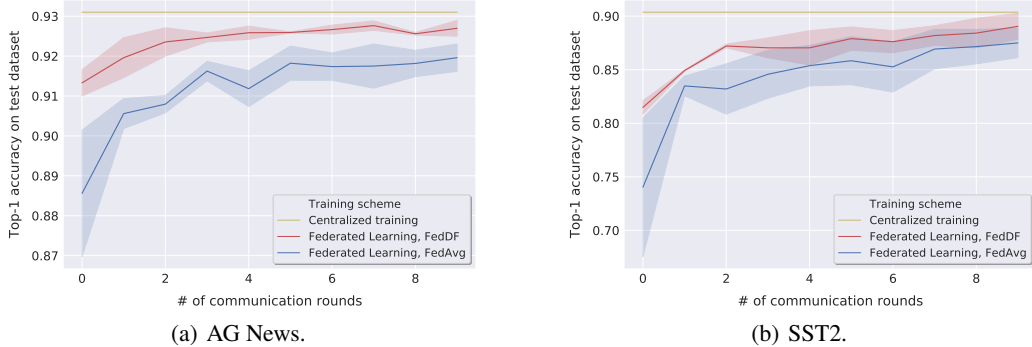

|              | (a) AG News. | (b) SST2. |

Figure 3: **Federated fine-tuning DistilBERT** on (a) AG News and (b) SST-2. For simplicity, we consider 10 clients with $C = 100\%$ participation ratio and $\alpha = 1$; the number of local training epochs per communication round (10 rounds in total) is set to 10 and 1 respectively. The $50\%$ of the original training dataset is used for the federated fine-tuning (for all methods) and the left $50\%$ is used as the unlabeled distillation dataset for FedDF.

Table 4: **Federated learning with low-precision models (1-bit binarized ResNet-8) on CIFAR-10**. For each communication round (100 in total), $40\%$ of the total 20 clients ($\alpha = 1$) are randomly selected.

| Local Epochs | ResNet-8-BN (FEDAVG) | ResNet-8-GN (FEDAVG) | ResNet-8-BN (**FedDF**) |
|---|---|---|---|
| 20 | $44.38 \pm 1.21$ | $\mathbf{59.70} \pm 1.65$ | $59.49 \pm 0.98$ |
| 40 | $43.91 \pm 3.26$ | $64.25 \pm 1.31$ | $\mathbf{65.49} \pm 0.74$ |
| 80 | $47.62 \pm 1.84$ | $65.99 \pm 1.29$ | $\mathbf{70.27} \pm 1.22$ |

**Extension to NLP tasks for federated fine-tuning of DistilBERT.** Fine-tuning a pre-trained transformer language model like BERT [13] yields SOTA results on various NLP benchmarks [73, 72]. DistilBERT [60] is a lighter version of BERT with only marginal quality loss on downstream tasks. As a proof of concept, in Figure 3 we consider federated fine-tuning of DistilBERT on non-i.i.d. local data ($\alpha = 1$, depicted in Figure 11). For both AG News and SST2 datasets, FedDF achieves significantly faster convergence than FEDAVG and consistently outperforms the latter.

### 4.3 Case Studies

**Federated learning for low-bit quantized models.** FL for the Internet of Things (IoT) involves edge devices with diverse hardware, e.g. different computational capacities. Network quantization is hence of great interest to FL by representing the activations/weights in low precision, with benefits of significantly reduced local computational footprints and communication costs. Table 4 examines the model fusion performance for binarized ResNet-8 [57, 30]. FedDF can be on par with or outperform FEDAVG by a noticeable margin, without introducing extra GN tuning overheads.

**Federated learning on heterogeneous systems.** Apart from non-i.i.d. local distributions, another major source of heterogeneity in FL systems manifests in neural architectures [41]. Figure 4 visualizes the training dynamics of FedDF and FEDAVG[9] in a heterogeneous system with three distinct architectures, i.e., ResNet-20, ResNet-32, and ShuffleNetV2. On CIFAR-10/100 and ImageNet, FedDF dominates FEDAVG on test accuracy in each communication round with much less variance. Each fused model exhibits marginal quality loss compared to the ensemble performance, which suggests unlabeled datasets from other domains are sufficient for model fusion. Besides, the gap between the fused model and the ensemble one widens when the training dataset contains a much larger number of classes[10] than that of the distillation dataset. For instance, the performance gap is negligible on CIFAR-10, whereas on ImageNet, the gap increases to around $6\%$. In Section 5, we study this underlying interaction between training data and unlabeled distillation data in detail.

## 5 Understanding FedDF

FedDF consists of two chief components: ensembling and knowledge distillation via out-of-domain data. In this section, we first investigate what affects the ensemble performance on the global distribution (test domain) through a generalization bound. We then provide empirical understanding of how different attributes of the out-of-domain distillation dataset affect the student performance on the global distribution.

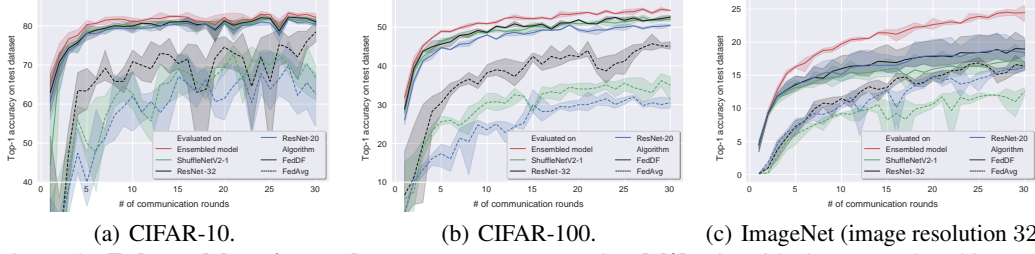

(a) CIFAR-10.  (b) CIFAR-100.  (c) ImageNet (image resolution 32).

Figure 4: **Federated learning on heterogeneous systems (model/data)**, with three neural architectures (ResNet-20, ResNet-32, ShuffleNetV2) and non-i.i.d. local data distribution ($\alpha = 1$). We consider 21 clients for CIFAR (client sampling ratio $C = 0.4$) and 150 clients for ImageNet ($C = 0.1$); different neural architectures are evenly distributed among clients. We train 80 local training epochs per communication round (total 30 rounds). CIFAR-100, STL-10, and STL-10 are used as the distillation datasets for CIFAR-10/100 and ImageNet training respectively. The *solid* lines show the results of FedDF for a given communication round, while *dashed* lines correspond to that of FEDAVG; *colors* indicate model architectures.

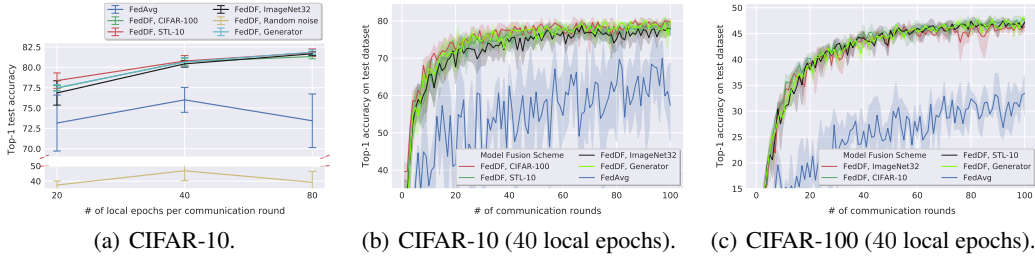

(a) CIFAR-10.  (b) CIFAR-10 (40 local epochs).  (c) CIFAR-100 (40 local epochs).

Figure 5: **The performance of FedDF on different distillation datasets**: random uniformly sampled noises, randomly generated images (from the generator), CIFAR, downsampled ImageNet32, and downsampled STL-10. We evaluate ResNet-8 on CIFAR for 20 clients, with $C = 0.4$, $\alpha = 1$ and 100 communication rounds.

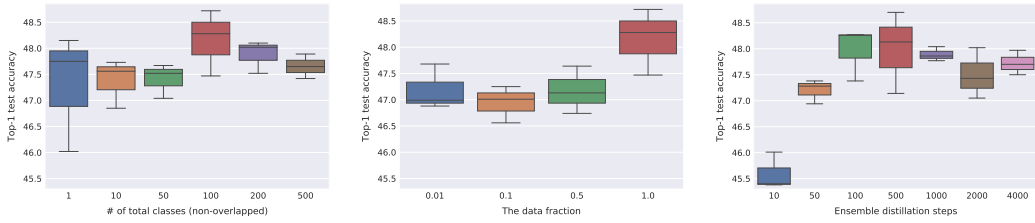

(a) The fusion performance of FedDF through unlabeled ImageNet, for different numbers of classes.  (b) The performance of FedDF via unlabeled ImageNet (100 classes), for different data fractions.  (c) The fusion performance of FedDF under different numbers of distillation steps.

Figure 6: **Understanding knowledge distillation behaviors of FedDF** on **# of classes** (6(a)), **sizes of the distillation dataset** (6(b)), and **# of distillation steps** (6(c)), for federated learning ResNet-8 on CIFAR-100, with $C = 0.4$, $\alpha = 1$ and 100 communication rounds (40 local epochs per round). ImageNet with image resolution 32 is considered as our base unlabeled dataset. For simplicity, only classes without overlap with CIFAR-100 classes are considered, in terms of the synonyms, hyponyms, or hypernyms of the class name.

**Generalization bound.** Theorem 5.1 provides insights into ensemble performance on the global distribution. Detailed description and derivations are deferred to Appendix D.

**Theorem 5.1** (informal)**.** *We denote the global distribution as $\mathcal{D}$, the $k$-th local distribution and its empirical distribution as $\mathcal{D}_k$ and $\hat{\mathcal{D}}_k$ respectively. The hypothesis $h \in \mathcal{H}$ learned on $\hat{\mathcal{D}}_k$ is denoted by $h_{\hat{\mathcal{D}}_k}$. The upper bound on the risk of the ensemble of $K$ local models on $\mathcal{D}$ mainly consists of 1) the empirical risk of a model trained on the global empirical distribution $\hat{\mathcal{D}} = \frac{1}{K}\sum_k \hat{\mathcal{D}}_k$, and 2) terms dependent on the distribution discrepancy between $\mathcal{D}_k$ and $\mathcal{D}$, with the probability $1 - \delta$:*

$$L_{\mathcal{D}}\left(\tfrac{1}{K}\sum_k h_{\hat{\mathcal{D}}_k}\right) \leq L_{\hat{\mathcal{D}}}(h_{\hat{\mathcal{D}}}) + \frac{1}{K}\sum_k \left(\frac{1}{2}d_{\mathcal{H}\Delta\mathcal{H}}(\mathcal{D}_k, \mathcal{D}) + \lambda_k\right) + \sqrt{\frac{\log\frac{2K}{\delta}}{2m}},$$

*where $d_{\mathcal{H}\Delta\mathcal{H}}$ measures the distribution discrepancy between two distributions [3], $m$ is the number of samples per local distribution, and $\lambda_k$ is the minimum of the combined loss $\mathcal{L}_{\mathcal{D}}(h) + \mathcal{L}_{\mathcal{D}_k}(h), \forall h \in \mathcal{H}$.*

The ensemble of the local models sets the performance upper bound for the later distilled model on the global distribution as shown in Figure 4. Theorem 5.1 shows that compared to a model trained

on the global empirical distribution (ideal centralized case), the performance of the ensemble on the global distribution is associated with the discrepancy between local distributions $\mathcal{D}_k$'s and the global distribution $\mathcal{D}$. Besides, the shift between the distillation and the global distribution determines the knowledge transfer quality between these two distributions and hence the test performance of the fused model. In the following, we empirically examine how the choice of distillation data distributions and the number of distillation steps influence the quality of ensemble knowledge distillation.

**Source, diversity and size of the distillation dataset.** The fusion in FedDF demonstrates remarkable consistency across a wide range of realistic data sources as shown in Figure 5, although an abrupt performance declination is encountered when the distillation data are sampled from a dramatically different manifold (e.g. random noise). Notably, synthetic data from the generator of a pre-trained GAN does not incur noticeable quality loss, opening up numerous possibilities for effective and efficient model fusion. Figure 6(a) illustrates that in general the diversity of the distillation data does not significantly impact the performance of ensemble distillation, though the optimal performance is achieved when two domains have a similar number of classes. Figure 6(b) shows the FedDF is not demanding on the distillation dataset size: even $1\%$ of data ($\sim 48\%$ of the local training dataset) can result in a reasonably good fusion performance.

**Distillation steps.** Figure 6(c) depicts the impact of distillation steps on fusion performance, where FedDF with a moderate number of the distillation steps is able to approach the optimal performance. For example, $100$ distillation steps in Figure 6(c), which corresponds to 5 local epochs of CIFAR-100 (partitioned by 20 clients), suffice to yield satisfactory performance. Thus FedDF introduces minor time-wise expense.

## Broader Impact

We believe that collaborative learning schemes such as federated learning are an important element towards enabling privacy-preserving training of ML models, as well as a better alignment of each individual's data ownership with the resulting utility from jointly trained machine learning models, especially in applications where data is user-provided and privacy sensitive [34, 55].
In addition to privacy, efficiency gains and lower resource requirements in distributed training reduce the environmental impact of training large machine learning models. The introduction of a practical and reliable distillation technique for heterogeneous models and for low-resource clients is a step towards more broadly enabling collaborative privacy-preserving and efficient decentralized learning.

## Acknowledgements

We acknowledge funding from SNSF grant 200021_175796, as well as a Google Focused Research Award.

## Footnotes

[2] GAN training is not involved in all stages of FL and cannot steal clients' data. Data generation is done by the (frozen) generator before the FL training by performing inference on random noise. Adversarially involving GAN's training during the FL training may cause the privacy issue, but it is beyond the scope of this paper.

[3] For instance, these data can be generated locally from identical generators with a controlled random state.

[4] The performance of FEDAVGM is coupled with local learning rate, local training epochs, and the number of communication rounds. The preprints [25, 26] consider small learning rate for at least 10k communication rounds; while we use much fewer communication rounds, which sometimes result in different observations.

[5] FEDMA does not support BN or residual connections, thus the comparison is only performed on VGG-9.

[6] Note the actual computation expense for distillation is determined by the product of the number of distillation steps and distillation mini-batch size (128 in all experiments), rather than the distillation dataset size.

[7] We include some preliminary results to illustrate the compatibility of FedDF in Table 8 (Appendix C.4.1).

[8] Techniques (e.g. Krum, Bulyan), can be adapted to further improve the robustness or defend against attacks.

[9] Model averaging is only performed among models with identical structures.

[10] # of classes is a proxy measurement for distribution shift; labels are not used in our distillation procedure.

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
