[Supplementary Material]

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

[11] The other methods use 40 local training epochs per whole model update. Given the fact of layer-wise training scheme in FEDMA, as well as the used 9-layer VGG (same as the one used in [74] and we are unable to

[14] The related preprints [41, 9] are closer to the second initialization scheme. They do not or cannot introduce the uniformly averaged model (on the server) into the federated learning pipeline; instead, they only utilize the averaged logits (on the same data) for each client's local training.

[15] The uniformly weighted hypothesis average in multi-source adaptation is equivalent to the ensemble of a list of models, by considering the output of each hypothesis/model.

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

# A  Detailed Related Work Discussion

**Prior work.**  We first comment on the two close approaches (FedMD and Cronus), in order to address 1) Distinctions between FedDF and prior work, 2) Privacy/Communication traffic concerns, 3) Omitted experiments on FedMD and Cronus.

- Distinctions between FedDF and prior work. As discussed in the related work, most SOTA FL methods directly manipulate received model parameters (e.g. FedAvg/FedAvgM/FedMA). To our best knowledge, FedMD and Cronus are the only two that utilize logits information (of neural nets) for FL. The distinctions from them are made below.
- Different objectives and evaluation metrics. Cronus is designed for robust FL under poisoning attack, whereas FedMD is for personalized FL. In contrast, FedDF is intended for on-server model aggregation (evaluation on the aggregated model), whereas neither FedMD nor Cronus aggregates the model on the server.
- Different Operations.
  1. FedDF, like FedAvg, *only* exchanges models between the server and clients, without transmitting input data. In contrast, FedMD and Cornus rely on exchanging public data logits. As FedAvg, FedDF can include privacy/security extensions and has the same communication cost per round.
  2. FedDF performs ensemble distillation with unlabeled data *on the server*. In contrast, FedMD/Cronus use averaged logits received from the server for *local client training*.
- Omitted experiments with FedMD/Cronus.
  1. FedMD requires to locally pre-train on the *labeled* public data, thus the model classifier necessitates an output dimension of # of public classes *plus* # of private classes (c.f. the output dimension of # of private classes in other FL methods). We cannot compare FedMD with FedDF with the same architecture (classifier) to ensure fairness.
  2. Cronus is shown to be consistently worse than FedAvg in normal FL (i.e. no attack case) in their Tab. IV & VI.
  3. Different objectives/metrics argued above. We thoroughly evaluated SOTA baselines with the same objective/metric.

**Contemporaneous work.**  We then detail some contemporaneous work, e.g. [68, 10, 81, 19]. [68] slightly extends FedMD by adding differential privacy. In [81], the server aggregates the synthetic data distilled from clients' private dataset, which in turn uses for one-shot on-server learning. He *et al* [19] improve FL for resource-constrained edge devices by combing FL with Split Learning (SL) and knowledge distillation: edge devices train compact feature extractor through local SGD and then synchronize extracted features and logits with the server, while the server (asynchronously) uses the latest received features and logits to train a much larger server-side CNN. The knowledge distillation is used on both the server and clients to improve the optimization quality.

FedDistill [10] is very similar to us, where it resorts to stochastic weight average-Gaussian (SWAG) [49] and the ensemble distillation is achieved via cyclical learning rate schedule with SWA [32]. In Table 7 below, we empirically compare our FedDF with this contemporaneous work (i.e. FedDistill).

# B  Algorithmic Description

Algorithm 2 below details a general training procedure on local clients. The local update step of FEDPROX corresponds to adding a proximal term (i.e. $\eta \frac{\partial \frac{\mu}{2} \left\| \mathbf{x}_t^k - \mathbf{x}_{t-1}^k \right\|_2^2}{\partial \mathbf{x}_t^k}$) to line 5.

Algorithm 3 illustrates the model fusion of FedDF for the FL system with heterogeneous model prototypes. The schematic diagram is presented in Figure 7. To perform model fusion in such heterogeneous scenarios, FedDF constructs several prototypical models on the server. Each prototype represents all clients with identical architecture/size/precision etc.

**Algorithm 2** Illustration of local client update in FEDAVG. The $K$ clients are indexed by $k$; $\mathcal{P}_k$ indicates the set of indexes of data points on client $k$, and $n_k = |\mathcal{P}_k|$. $E$ is the number of local epochs, and $\eta$ is the learning rate. $\ell$ evaluates the loss on model weights for a mini-batch of an arbitrary size.

1: **procedure** CLIENT-LOCALUPDATE$(k, \mathbf{x}_{t-1}^k)$
2:    Client $k$ receives $\mathbf{x}_{t-1}^k$ from server and copies it as $\mathbf{x}_t^k$
3:    **for** each local epoch $i$ from 1 to $E$ **do**
4:      **for** mini-batch $b \subset \mathcal{P}_k$ **do**
5:        $\mathbf{x}_t^k \leftarrow \mathbf{x}_t^k - \eta \frac{\partial \ell(\mathbf{x}_t^k; b)}{\partial \mathbf{x}_t^k}$        ▷ can be arbitary optimizers (e.g. Adam)
6:    **return** $\mathbf{x}_t^k$ to server

---

**Algorithm 3** Illustration of FedDF for heterogeneous FL systems. The $K$ clients are indexed by $k$, and $n_k$ indicates the number of data points for the $k$-th client. The number of communication rounds is $T$, and $C$ controls the client participation ratio per communication round. The number of total iterations used for model fusion is denoted as $N$. The distinct model prototype set $\mathcal{P}$ has $p$ model prototypes, with each initialized as $\mathbf{x}_0^P$.

1: **procedure** SERVER
2:    initialize HashMap $\mathcal{M}$: map each model prototype $P$ to its weights $\mathbf{x}_0^P$.
3:    initialize HashMap $\mathcal{C}$: map each client to its model prototype.
4:    initialize HashMap $\tilde{\mathcal{C}}$: map each model prototype to the associated clients.
5:    **for** each communication round $t = 1, \ldots, T$ **do**
6:      $\mathcal{S}_t \leftarrow$ a random subset ($C$ fraction) of the $K$ clients
7:      **for** each client $k \in \mathcal{S}_t$ **in parallel do**
8:        $\hat{\mathbf{x}}_t^k \leftarrow$ Client-LocalUpdate$(k, \mathcal{M}[\mathcal{C}[k]])$        ▷ detailed in Algorithm 2.
9:      **for** each prototype $P \in \mathcal{P}$ **in parallel do**
10:        initialize the client set $\mathcal{S}_t^P$ with model prototype $P$, where $\mathcal{S}_t^P \leftarrow \tilde{\mathcal{C}}[P] \cap \mathcal{S}_t$
11:        initialize for model fusion $\mathbf{x}_{t,0}^P \leftarrow \sum_{k \in \mathcal{S}_t^P} \frac{n_k}{\sum_{k \in \mathcal{S}_t^P} n_k} \hat{\mathbf{x}}_t^k$
12:        **for** $j$ in $\{1, \ldots, N\}$ **do**
13:          sample $\mathbf{d}$, from e.g. (1) an unlabeled dataset, (2) a generator
14:          use ensemble of $\{\hat{\mathbf{x}}_t^k\}_{k \in \mathcal{S}_t}$ to update server student $\mathbf{x}_{t,j}^P$ through AVGLOGITS
15:        $\mathcal{M}[P] \leftarrow \mathbf{x}_{t,N}^P$
16:    **return** $\mathcal{M}$

Figure 7: **The schematic diagram for heterogeneous model fusion.** We use dotted lines to indicate model parameter averaging FL methods such as FEDAVG. We could notice the architectural/precision discrepancy invalidates these methods in heterogeneous FL systems. However, FedDF could aggregate knowledge from all available models without hindrance.

# C Additional Experimental Setup and Evaluations

## C.1 Detailed Description for Toy Example (Figure 1)

Figure 8 provides a detailed illustration of the limitation in FEDAVG.

Figure 8: **The limitation of FEDAVG.** We consider a toy example of a 3-class classification task with a 3-layer MLP, and display the decision boundaries (probabilities over RGB channels) on the input space. We illustrate the used datasets in the **top** row; the distillation dataset consists of 60 data points, with each uniformly sampled from the range of $(-3, 3)$. In the **bottom** row, the left two figures consider the individually trained local models. The right three figures evaluate aggregated models and the global data distribution; the averaged model (FEDAVG) results in much blurred decision boundaries.

## C.2 Detailed Experiment Setup

**The detailed hyperparameter tuning procedure.** The tuning procedure of hyperparameters ensures that the best hyperparameter lies in the middle of our search grids; otherwise, we extend our search grid. The initial search grid of learning rate is $\{1.5, 1, 0.5, 0.1, 0.05, 0.01\}$. The initial search grid of proximal factor in FEDPROX is $\{0.001, 0.01, 0.1, 1\}$. The initial search grid of momentum factor $\beta$ in FEDAVGM is $\{0.1, 0.2, 0.3, 0.4\}$; the update scheme of FEDAVGM follows $\Delta\mathbf{v} := \beta\mathbf{v} + \Delta\mathbf{x}$ ; $\mathbf{x} := \mathbf{x} - \Delta\mathbf{v}$, where $\Delta\mathbf{x}$ is the model difference between the updated local model and the sent global model, for previous communication round.

Unless mentioned (i.e. Table 1), otherwise the learning rate is set to $0.1$ for ResNet like architectures (e.g. ResNet-8, ResNet-20, ResNet-32, ShuffleNetV2), $0.05$ for VGG and $1e-5$ for DistilBERT. When comparing with other methods, e.g. FEDPROX, FEDAVGM, we always tune their corresponding hyperparameters (e.g. proximal factor in FEDPROX and momentum factor in FEDAVGM).

**Experiment details of FEDMA.** We detail our attempts of reproducing FEDMA experiments on VGG-9 with CIFAR-10 in this section. We clone their codebase from GitHub and add functionality to sample clients after synchronizing the whole model.

Different from other methods evaluated in the paper, FEDMA uses a layer-wise local training scheme. For each round of the local training, the involved clients only update the model parameters from one specific layer onwards, while the already matched layers are frozen. The fusion (matching) is only performed on the chosen layer. Such a layer is gradually chosen from the bottom layer to the top layer, following a bottom-up fashion [74]. One complete model update cycle of FEDMA requires more frequent (but slightly cheaper) communication, which is equivalent to the number of layers in the neural network.

In our experiments of FEDMA, the number of local training epochs is 5 epochs per layer (45 epochs per model update), which is slightly larger than 40 epochs used by other methods. We ensure a similar[11] number of model updates in terms of the whole model. We consider global-wise learning

rate, different from the layer-wise one in Wang et al. [74]. We also turn off the momentum and weight decay during the local training for a consistent evaluation. The implementation of VGG-9 follows `https://github.com/kuangliu/pytorch-cifar/`.

**The detailed experimental setup for FedDF (low-bit quantized models).** FedDF increases the feasibility of robust model fusion in FL for binarized ResNet-8. As stated in Table 4 (Section 4.3), we employ the "Straight-through estimator" [4, 21, 29, 30] or the "error-feedback" [45] to simulate the on-device local training of the binarized ResNet-8. For each communication round, the server of the FL system will receive locally trained and binarized ResNet-8 from activated clients. The server will then distill the knowledge of these low-precision models to a full-precision one[12] and broadcast to newly activated clients for the next communication round. For the sake of simplicity, the case study demonstrated in the paper only considers reducing the communication cost (from clients to the server), and the local computational cost; a thorough investigation on how to perform a communication-efficient and memory-efficient FL is left as future work.

**The synthetic formulation of non-i.i.d. client data.** Assume every client training example is drawn independently with class labels following a categorical distribution over $M$ classes parameterized by a vector $\mathbf{q}$ ($q_i \geq 0, i \in [1, M]$ and $\|\mathbf{q}\|_1 = 1$). Following the partition scheme introduced and used in [78, 25][13], to synthesize client non-i.i.d. local data distributions, we draw $\alpha \sim \mathrm{Dir}(\alpha \mathbf{p})$ from a Dirichlet distribution, where $\mathbf{p}$ characterizes a prior class distribution over $M$ classes, and $\alpha > 0$ is a concentration parameter controlling the identicalness among clients. With $\alpha \to \infty$, all clients have identical distributions to the prior; with $\alpha \to 0$, each client holds examples from only one random class.

To better understand the local data distribution for the datasets we considered in the experiments, we visualize the partition results of CIFAR-10 and CIFAR-100 on $\alpha = \{0.01, 0.1, 0.5, 1, 100\}$ for 20 clients, in Figure 9 and Figure 10, respectively.

In Figure 11 we visualize the partitioned local data on 10 clients with $\alpha = 1$, for AG News and SST-2.

(a) $\alpha = 100$      (b) $\alpha = 1$      (c) $\alpha = 0.5$

(d) $\alpha = 0.1$            (e) $\alpha = 0.01$

Figure 9: Classes allocated to each client at different Dirichlet distribution alpha values, for CIFAR-10 with 20 clients. The size of each dot reflects the magnitude of the samples number.

## C.3 Some Empirical Understanding of FEDAVG

Figure 12 reviews the general behaviors of FEDAVG under different non-iid degrees of local data, different local data sizes, different numbers of local epochs per communication round, as well as the learning rate schedule during the local training. Since we cannot observe the benefits of decaying the

---

adapt their code to other architectures due to their hard-coded architecture manipulations), we decide to slightly increase the number of local epochs per layer for FEDMA.

[12] The training of the binarized network requires to maintain a full-precision model [29, 30, 45] for model update (quantized/pruned model is used during the backward pass).

[13] We heavily borrowed the partition description of [25] for the completeness of the paper.

Figure 10: Classes allocated to each client at different Dirichlet distribution alpha values, for CIFAR-100 with 20 clients. The size of each dot reflects the magnitude of the samples number.

Figure 11: Classes allocated to each client at Dirichlet distribution $\alpha = 1$, for AG News and SST2 datasets with 10 clients. The size of each dot reflects the magnitude of the samples number.

learning rate during the local training phase, we turn off the learning rate decay for the experiments in the main text.

In Figure 13, we visualize the learning curves of training ResNet-8 on CIFAR-10 with different normalization techniques. The numerical results correspond to Table 2 in the main text.

Figure 12: **The ablation study of FEDAVG for different # of local epochs and learning rate schedules**, for standard federated learning on CIFAR-10 with ResNet-8. For each communication round (100 in total), 40% of the total 20 clients are randomly selected. We use $\alpha$ to synthetically control the non-iid degree of the local data, as in [78, 25]. The smaller $\alpha$, the larger discrepancy between local data distributions ($\alpha = 100$ mimics identical local data distributions). We report the top-1 accuracy (on three different seeds) on the test dataset.

Figure 13: The impact of different normalization techniques, i.e., Batch Normalization (BN), Group Normalization (GN), for federated learning on CIFAR-10 with ResNet-8 with $\alpha = 1$. For each communication round (100 in total), $40\%$ of the total 20 clients are randomly selected for 40 local epochs.

## C.4 The Advantages of FedDF

### C.4.1 Ablation Study

**The Importance of the Model Initialization in FedDF.** We empirically study the importance of the initialization (before performing ensemble distillation) in FedDF. Table 5 demonstrates the performance difference of FedDF for two different model initialization schemes: 1) "from average", where the uniformly averaged model from this communication round is used as the initial model (i.e. the default design choice of FedDF as illustrated in Algorithm 1 and Algorithm 3); and 2) "from previous", where we initialize the model for ensemble distillation by utilizing the fusion result of FedDF from the previous communication round. The noticeable performance differences illustrated in Table 5 identify the importance of using the uniformly averaged model[14] (from the current communication round) as a starting model for better ensemble distillation.

Table 5: **Understanding the importance of model initialization in FedDF**, on CIFAR-10 with ResNet-8. For each communication round (100 in total), $40\%$ of the total 20 clients are randomly selected. The scheme "from average" indicates initializing the model for ensemble distillation from the uniformly averaged model of this communication round; while the scheme "from previous" instead uses the fused model from the previous communication round as the starting point. We report the top-1 accuracy (on three different seeds) on the test dataset.

| local training epochs | $\alpha = 1$ | | $\alpha = 0.1$ | |
|---|---|---|---|---|
| | from average | from previous | from average | from previous |
| 40 | $80.43 \pm 0.37$ | $74.13 \pm 0.91$ | $71.84 \pm 0.86$ | $62.94 \pm 1.12$ |
| 80 | $81.17 \pm 0.53$ | $76.37 \pm 0.60$ | $74.73 \pm 0.65$ | $67.88 \pm 0.90$ |

**The performance gain in FedDF.** To distinguish the benefits of FedDF from the small learning rate (during the local training) or Adam optimizer (used for ensemble distillation in FedDF), we report the results of using Adam (lr=1e-3) for both local training and model fusion (over three seeds), on CIFAR-10 with ResNet-8, in Table 6. Improving the local training through Adam might help Federated Learning but the benefit vanishes with higher data heterogeneity (e.g. $\alpha = 0.1$). Performance gain from FedDF is robust to data heterogeneity and also orthogonal to effects of learning rates and Adam.
Table 7 examines the effect of different optimization schemes on the quality of ensemble distillation. We can witness that with two extra hyper-parameters (sampling scale for SWAG and the number of models to be sampled), SWAG can slightly improve the distillation performance. In contrast, we use Adam with default hyper-parameters as our design choice in FedDF: it demonstrates similar performance (compared to the choice of SWAG) with trivial tuning overhead.

**The compatibility of FedDF with other methods.** Table 8 justifies the compatibility of FedDF. Our empirical results demonstrate a significant performance gain of FedDF over the FEDAVG, even

Table 6: **Understanding the impact of local training quality**, on CIFAR-10 with ResNet-8. For each communication round (100 in total), 40% of the total 20 clients are randomly selected for 40 local epochs. We report the top-1 accuracy (on three different seeds) on the test dataset.

| local client training scheme | $\alpha=1$ | | $\alpha=0.1$ | |
|---|---|---|---|---|
| | FedDF | FEDAVG | FedDF | FEDAVG |
| SGD | 80.27 | 72.73 | 71.52 | 62.44 |
| Adam | 83.32 | 78.13 | 72.58 | 62.53 |

Table 7: **On the impact of using different optimizers for ensemble distillation** in FedDF, on CIFAR-10 with ResNet-8. For each communication round (100 in total), 40% of the total 20 clients are randomly selected for 40 local epochs. We report the top-1 accuracy (on three different seeds) on the test dataset. "SGD" uses the same learning rate scheduler as our "Adam" choice (i.e. cosine annealing), and with fine-tuned initial learning rate. "SWAG" refers to the mechanism to form an approximated posterior distribution [49] where more models can be sampled from, and [10] further propose to use SWAG on the received client models for better ensemble distillation; our default design resorts to directly averaged logits from received local clients with Adam optimizer. To ensure a fair comparison, we use the same distillation dataset as in FedDF (i.e., CIFAR-100) for "SWAG" [10]. We fine-tune other hyper-parameters in "SWAG": we use all received client models and 10 sampled models from Gaussian distribution (as suggested in [10]) for the ensemble distillation.

| optimizer used on the server | $\alpha=1$ | | $\alpha=0.1$ | |
|---|---|---|---|---|
| | FedDF | FEDAVG | FedDF | FEDAVG |
| SGD | 76.68 | 72.73 | 57.33 | 62.44 |
| Adam (our default design) | 80.27 | 72.73 | 71.52 | 62.44 |
| SWAG [49, 10] | 80.84 | 72.73 | 72.40 | 62.44 |

in the case of using local proximal regularizer to avoid catastrophically over-fitting the heterogeneous local data, which reduces the diversity of local models that FedDF benefits from.

Table 8: **The compatibility of FedDF with other training schemes**, on CIFAR-10 with ResNet-8. For each communication round (100 in total), 40% of the total 20 clients are randomly selected for 40 local epochs. We consider the fine-tuned proximal penalty from FedDF. We report the top-1 accuracy (on three different seeds) on the test dataset.

| local client training scheme | $\alpha=1$ | | $\alpha=0.1$ | |
|---|---|---|---|---|
| | FedDF | FEDAVG | FedDF | FEDAVG |
| SGD | 80.27 | 72.73 | 71.52 | 62.44 |
| SGD + proximal penalty | 80.56 | 76.11 | 71.64 | 62.53 |

### C.4.2 Comparison with FEDAVG

Figure 14 complements Figure 2 in the main text and presents a thorough comparison between FEDAVG and FedDF, for a variety of different local training epochs, data fractions, non-i.i.d. degrees. The detailed learning curves of the cases in this figure are visualized in Figure 15, Figure 16, and Figure 17.

(a) $\alpha=100$.  (b) $\alpha=1$.  (c) $\alpha=0.01$.

Figure 14: The **test performance** of **FedDF** and **FEDAVG** on **CIFAR-10** with **ResNet-8**, for different local data non-iid degrees $\alpha$, data fractions, and # of local epochs per communication round. For each communication round (100 in total), 40% of the total 20 clients are randomly selected. We report the top-1 accuracy (on three different seeds) on the test dataset. This Figure complements Figure 2.

(a) The learning behaviors of FedDF and FEDAVG. We evaluate different # of local epochs on 100% local data.

(b) The fused model performance before (i.e. line 6 in Algorithm 1) and after FedDF (i.e. line 10 in Algorithm 1). We evaluate different # of local epochs on 100% local data.

(c) The learning behaviors of FedDF and FEDAVG. We evaluate different # of local epochs on 50% local data.

(d) The fused model performance before (i.e. line 6 in Algorithm 1) and after FedDF (i.e. line 10 in Algorithm 1). We evaluate different # of local epochs on 50% local data.

Figure 15: **Understanding the learning behaviors of FedDF** on CIFAR-10 with ResNet-8 for $\alpha = 100$. For each communication round (100 in total), 40% of the total 20 clients are randomly selected. We report the top-1 accuracy (on three different seeds) on the test dataset.

(a) The learning behaviors of FedDF and FEDAVG. We evaluate different # of local epochs on 100% local data.

(b) The fused model performance before (i.e. line 6 in Algorithm 1) and after FedDF (i.e. line 10 in Algorithm 1). We evaluate different # of local epochs on 100% local data.

(c) The learning behaviors of FedDF and FEDAVG. We evaluate different # of local epochs on 50% local data.

(d) The fused model performance before (i.e. line 6 in Algorithm 1) and after FedDF (i.e. line 10 in Algorithm 1). We evaluate different # of local epochs on 50% local data.

Figure 16: **Understanding the learning behaviors of FedDF** on CIFAR-10 with ResNet-8 for $\alpha = 1$. For each communication round (100 in total), $40\%$ of the total 20 clients are randomly selected. We report the top-1 accuracy (on three different seeds) on the test dataset.

(a) The learning behaviors of FedDF and FEDAVG. We evaluate different # of local epochs on $100\%$ local data.

(b) The fused model performance before (i.e. line 6 in Algorithm 1) and after FedDF (i.e. line 10 in Algorithm 1). We evaluate different # of local epochs on $100\%$ local data.

(c) The learning behaviors of FedDF and FEDAVG. We evaluate different # of local epochs on $50\%$ local data.

(d) The fused model performance before (i.e. line 6 in Algorithm 1) and after FedDF (i.e. line 10 in Algorithm 1). We evaluate different # of local epochs on $50\%$ local data.

Figure 17: **Understanding the learning behaviors of FedDF** on CIFAR-10 with ResNet-8 for $\alpha = 0.01$. For each communication round (100 in total), $40\%$ of the total 20 clients are randomly selected. We report the top-1 accuracy (on three different seeds) on the test dataset.

# D   Details on Generalization Bounds

The derivation of the generalization bound starts from the following notations. In FL, each client has access to its own data distribution $\mathcal{D}_i$ over domain $\Xi := \mathcal{X} \times \mathcal{Y}$, where $\mathcal{X} \in \mathbb{R}^d$ is the input space and $\mathcal{Y}$ is the output space. The global distribution on the server is denoted as $\mathcal{D}$. For the empirical distribution by the given dataset, we assume that each local model has access to an equal amount $(m)$ of local data. Thus, each local empirical distribution has equal contribution to the global empirical distribution: $\hat{\mathcal{D}} = \frac{1}{K} \sum_{k=1}^{K} \hat{\mathcal{D}}_k$, where $\hat{\mathcal{D}}_k$ denotes the empirical distribution from client $k$.

For our analysis we assume a binary classification task, with hypothesis $h$ as a function $h : \mathcal{X} \rightarrow \{0, 1\}$. The loss function of the task is defined as $\ell(h(\mathbf{x}), y) = |\hat{y} - y|$, where $\hat{y} := h(\mathbf{x})$. Note that $\ell(\hat{y}, y)$ is convex with respect to $\hat{y}$. We denote $\arg\min_{h \in \mathcal{H}} L_{\hat{\mathcal{D}}}(h)$ by $h_{\hat{\mathcal{D}}}$.

The theorem below leverages the domain measurement tools developed in multi-domain learning theory [3] and provides insights for the generalization bound of the ensemble[15] of local models (trained on local empirical distribution $\hat{\mathcal{D}}_i$).

**Theorem D.1.** *The difference between $L_{\mathcal{D}}(\frac{1}{K} \sum_k h_{\hat{\mathcal{D}}_k})$ and $L_{\hat{\mathcal{D}}}(h_{\hat{\mathcal{D}}})$, i.e., the distance between the risk of our "ensembled" model in FedDF and the empirical risk of the "virtual ERM" with access to all local data, can be bounded with probability at least $1 - \delta$:*

$$L_{\mathcal{D}}\left( \frac{1}{K} \sum_k h_{\hat{\mathcal{D}}_k} \right) \leq L_{\hat{\mathcal{D}}}(h_{\hat{\mathcal{D}}}) + \sqrt{\frac{\log \frac{2K}{\delta}}{2m}} + \frac{1}{K} \sum_k \left( \frac{1}{2} d_{\mathcal{H} \Delta \mathcal{H}}(\mathcal{D}_k, \mathcal{D}) + \lambda_k \right),$$

*where $\hat{\mathcal{D}} = \frac{1}{K} \sum_k \hat{\mathcal{D}}_k$, $d_{\mathcal{H} \Delta \mathcal{H}}$ measures the domain discrepancy between two distributions [3], and $\lambda_k = \inf_{h \in \mathcal{H}} (\mathcal{L}_{\mathcal{D}}(h) + \mathcal{L}_{\mathcal{D}_k}(h))$.*

**Remark D.2.** *Theorem D.1 shows that, the upper bound on the risk of the ensemble of K local models on $\mathcal{D}$ mainly consists of 1) the empirical risk of a model trained on the global empirical distribution $\hat{\mathcal{D}} = \frac{1}{K} \sum_k \hat{\mathcal{D}}_k$, and 2) terms dependent on the distribution discrepancy between $\mathcal{D}_k$ and $\mathcal{D}$.*

The ensemble of the local models sets the performance upper bound for the later distilled model on the test domain as shown in Figure 4. Theorem 5.1 shows that compared to a model trained on aggregated local data (ideal case), the performance of an ensemble model on the test distribution is affected by the domain discrepancy between local distributions $\mathcal{D}_k$'s and the test distribution $\mathcal{D}$. The shift between the distillation and the test distribution determines the knowledge transfer quality between these two distributions and hence the test performance of the fused model. Through the lens of the domain adaptation theory [3], we can better spot the potential influence/limiting factors on our ensemble distillation procedure.

**Remark D.3.** *In the area of multiple-source adaptation, [50, 23] point out that the standard convex combinations of the source hypotheses may perform poorly on the test distribution. They propose combinations with weights derived from source distributions. However, FL scenarios require the server only access local models without any further local information. Thus we choose to uniformly average over local hypotheses as our global hypothesis. A privacy-preserved local distribution estimation is left for future work.*

## D.1   Proof for Generalization Bounds

**Theorem D.4** (Domain adaptation [3]). *Considering the distributions $\mathcal{D}_S$ and $\mathcal{D}_T$, for every $h \in \mathcal{H}$ and any $\delta \in (0, 1)$, with probability at least $1 - \delta$ (over the choice of the samples), there exists:*

$$L_{\mathcal{D}_T}(h) \leq L_{\mathcal{D}_S}(h) + \frac{1}{2} d_{\mathcal{H} \Delta \mathcal{H}}(\mathcal{D}_S, \mathcal{D}_T) + \lambda, \tag{1}$$

*where $\lambda = L_{\mathcal{D}_S}(h^\star) + L_{\mathcal{D}_T}(h^\star)$. $h^\star := \arg\min_{h \in \mathcal{H}} L_{\mathcal{D}_S}(h) + L_{\mathcal{D}_T}(h)$ corresponds to* ideal joint hypothesis *that minimizes the combined error.*

*Proof of Theorem D.1.* We start from the risk of our "ensembled" model $L_{\mathcal{D}}(\frac{1}{K} \sum_k h_{\hat{\mathcal{D}}_k})$ and derive a series of upper bounds.

**Considering the distance between** $L_\mathcal{D}(\frac{1}{K}\sum_k h_{\hat{\mathcal{D}}_k})$ **and** $L_{\hat{\mathcal{D}}}(h_{\hat{\mathcal{D}}})$**.** By convexity of $\ell$ and Jensen inequality, we have

$$L_\mathcal{D}(\frac{1}{K}\sum_k h_{\hat{\mathcal{D}}_k}) \le \frac{1}{K}\sum_k L_\mathcal{D}(h_{\hat{\mathcal{D}}_k}) \,. \tag{2}$$

Using the domain adaptation theory in Theorem D.4, we transfer from domain $\mathcal{D}$ to $\mathcal{D}_k$,

$$L_\mathcal{D}(h_{\hat{\mathcal{D}}_k}) \le L_{\mathcal{D}_k}(h_{\hat{\mathcal{D}}_k}) + \frac{1}{2}d_{\mathcal{H}\Delta\mathcal{H}}(\mathcal{D}_k, \mathcal{D}) + \lambda_k \,, \tag{3}$$

where $\lambda_k := \mathcal{L}_\mathcal{D}(h^\star) + \mathcal{L}_{\mathcal{D}_k}(h^\star)$ and $h^\star := \arg\min_{h\in\mathcal{H}} \mathcal{L}_\mathcal{D}(h) + \mathcal{L}_{\mathcal{D}_k}(h)$.
We can bound the risk with its empirical counterpart through Hoeffding inequality. A simple application of the Hoeffding's inequality gives

$$\Pr\left[\left|L_{\mathcal{D}_k}(h_{\hat{\mathcal{D}}_k}) - L_{\hat{\mathcal{D}}_k}(h_{\hat{\mathcal{D}}_k})\right| \ge \epsilon\right] \le 2\exp\frac{-2m^2\epsilon^2}{\sum_{j=1}^m (b-a)^2} \,,$$

where $[a,b]$ is the range of loss function. In our case, the loss function is bounded in $[0,1]$ so $(b-a)^2 \le 1$, thereby, with probability at least $1 - \frac{\delta}{K}$, over the draw of $m$ i.i.d. samples $S_k$ from $\mathcal{D}_k$,

$$L_{\mathcal{D}_k}(h_{\hat{\mathcal{D}}_k}) \le L_{\hat{\mathcal{D}}_k}(h_{\hat{\mathcal{D}}_k}) + \sqrt{\frac{\log\frac{2}{\frac{\delta}{K}}}{2m}} \,, \tag{4}$$

Thus for $K$ sources, we have

$$\Pr_{S_1\sim\mathcal{D}_1^m,\dots,S_K\sim\mathcal{D}_K^m}\left[\bigcap_{k=1}^K\left\{L_{\mathcal{D}_k}(h_{\hat{\mathcal{D}}_k}) \le L_{\hat{\mathcal{D}}_k}(h_{\hat{\mathcal{D}}_k}) + \sqrt{\frac{\log\frac{2}{\frac{\delta}{K}}}{2m}}\right\}\right]$$

$$= 1 - \Pr_{S_1\sim\mathcal{D}_1^m,\dots,S_K\sim\mathcal{D}_K^m}\left[\bigcup_{k=1}^K\left\{L_{\mathcal{D}_k}(h_{\hat{\mathcal{D}}_k}) \ge L_{\hat{\mathcal{D}}_k}(h_{\hat{\mathcal{D}}_k}) + \sqrt{\frac{\log\frac{2}{\frac{\delta}{K}}}{2m}}\right\}\right] \tag{5}$$

$$\ge 1 - \sum_{k=1}^K \Pr_{S_1\sim\mathcal{D}_1^m,\dots,S_K\sim\mathcal{D}_K^m}\left[\left\{L_{\mathcal{D}_k}(h_{\hat{\mathcal{D}}_k}) \ge L_{\hat{\mathcal{D}}_k}(h_{\hat{\mathcal{D}}_k}) + \sqrt{\frac{\log\frac{2K}{\delta}}{2m}}\right\}\right]$$

$$\ge 1 - \delta \,.$$

Based on the definition of ERM, we have $L_{\hat{\mathcal{D}}_k}(h_{\hat{\mathcal{D}}_k}) \le L_{\hat{\mathcal{D}}_k}(h_{\hat{\mathcal{D}}})$, where $h_{\hat{\mathcal{D}}}$ corresponds to the classifier trained with data from all workers. By using the definition of $\hat{\mathcal{D}}$ ($\hat{\mathcal{D}} = \frac{1}{K}\sum_k \hat{\mathcal{D}}_k$) and the linearity of expectation, we have

$$\frac{1}{K}\sum_k L_{\hat{\mathcal{D}}_k}(h_{\hat{\mathcal{D}}_k}) \le \frac{1}{K}\sum_k L_{\hat{\mathcal{D}}_k}(h_{\hat{\mathcal{D}}}) = L_{\hat{\mathcal{D}}}(h_{\hat{\mathcal{D}}}) \,. \tag{6}$$

Putting these equations together, we have with probability of at least $1-\delta$ over $S_1 \sim \mathcal{D}_1^m,\dots,S_K \sim \mathcal{D}_K^m$ that

$$L_\mathcal{D}(\frac{1}{K}\sum_k h_{\hat{\mathcal{D}}_k}) \le \frac{1}{K}\sum_k L_\mathcal{D}(h_{\hat{\mathcal{D}}_k})$$

$$\le \frac{1}{K}\sum_k \left(L_{\mathcal{D}_k}(h_{\hat{\mathcal{D}}_k}) + \frac{1}{2}d_{\mathcal{H}\Delta\mathcal{H}}(\mathcal{D}_k, \mathcal{D}) + \lambda_k\right)$$

$$\le \frac{1}{K}\sum_k \left(L_{\hat{\mathcal{D}}_k}(h_{\hat{\mathcal{D}}_k}) + \sqrt{\frac{\log\frac{2K}{\delta}}{2m}} + \frac{1}{2}d_{\mathcal{H}\Delta\mathcal{H}}(\mathcal{D}_k, \mathcal{D}) + \lambda_k\right)$$

$$\le \frac{1}{K}\sum_k L_{\hat{\mathcal{D}}_k}(h_{\hat{\mathcal{D}}_k}) + \sqrt{\frac{\log\frac{2K}{\delta}}{2m}} + \frac{1}{K}\sum_k \left(\frac{1}{2}d_{\mathcal{H}\Delta\mathcal{H}}(\mathcal{D}_k, \mathcal{D}) + \lambda_k\right)$$

$$\le L_{\hat{\mathcal{D}}}(h_{\hat{\mathcal{D}}}) + \sqrt{\frac{\log\frac{2K}{\delta}}{2m}} + \frac{1}{K}\sum_k \left(\frac{1}{2}d_{\mathcal{H}\Delta\mathcal{H}}(\mathcal{D}_k, \mathcal{D}) + \lambda_k\right) \,,$$

where $\lambda_k = \inf_{h\in\mathcal{H}}\left(\mathcal{L}_\mathcal{D}(h) + \mathcal{L}_{\mathcal{D}_k}(h)\right)$.

$\square$