[Reviews · NeurIPS 2020]

Review 1

Summary and Contributions: This paper investigates a novel aggregation schemes to handle heterogeneous models and heterogeneous data problems across the clients. In particular, it ensemble various model architectures into group that has homogenous models in each group, and then use the ensembled model of each group to distil a student model with the assistance of an unlabelled dataset.

Strengths: (1) The targeting problem setting is really useful and important for federated learning. (2) The experimental study is strong enough to support the claims.

Weaknesses: (1) The proposed method requires the existance of an unalbelled dataset or data generator. It is a strong assumption in real-world use. It is also will be a challenge task if we consdier the non-iid distribution across clients. (2) The use of GAN to generate unlabelled data need to carefully designed. Because most of gradient-based privacy attack is based on the same technology, thus GAN could be used to generate unlabelled data and also steal user’s private data. It will cause many privacy concerns for the use of federated learning framework. (3) It would be interested to see the discussion of large-scale ensemble learning. I am curios to konw whethere there is any performance different for an ensemble model with tens of base models and thousands of base models. (4) The content needs to be re-organized. The paper’s contents should be self-contained; however, I have to read the version with supplementary for getting the missed information in 8-page submission.

Correctness: The method is correct.

Clarity: The organisation of the content need to be imporved.

Relation to Prior Work: The paper need to discuss other knowledge distillation-based federated learning methods [1],[2] and collaborative training of knowledge distillation [3] [4]. [1] Communication-Efficient On-Device Machine Learning: Federated Distillation and Augmentation under Non-IID Private Data [2] FedMD: Heterogeneous Federated Learning via Model Distillation [3] Online knowledge distillation via collaborative learning [4] Large scale distributed neural network training through online distillation

Reproducibility: Yes

Additional Feedback:


Review 2

Summary and Contributions: This paper proposed a model fusion federated learning method FedMD, which use ensemble distillation for robust model fusion. FedDF can allow for heterogeneous client models and data, and the fusion server model training with fewer communication rounds than FEDAVG, FEDPROX, FEDAVGM, FEDMA on CV and NLP tasks.

Strengths: (1)The proposed FedDF can considerably reduce the communication round for the server model training by leveraging ensemble distillation method and the unlabeled data. (2)The experiments in several different setting (heterogeneous data and low-bit quantized model) are conducted. A theoretical analysis about the factor of server model performance is introduced.

Weaknesses: (1) In Ln173, all competing methods have strong difficulties with increasing data heterogeneity, However, in tab 1, there not existing comparison of FedMD and Cronus, which have considerable similarity with FedDF. (2) In Ln117, due to the privacy cost, FedDF adding differential privacy are not practical, the FedDF method reduces the number of communication rounds, but the loss of AVGLOGITS requires the logit outputs of the client model, hence the server need transmit the input data, which lead to the communication traffic is large and limits the application of FedDF (3) The number of communication rounds is reduced, but whether the accuracy of the model is degraded is not known, no experiments have been done, for example, the medical scenario requires a very high degree of accuracy, but tab1 only shows the model to achieve 80% to 75% accuracy. (4) In Ln216, there should have comparison with FedMD, which are designed for heterogeneous systems. (5) Ln196 received model refer to?

Correctness: There are many errors

Clarity: see the weaknesses.

Relation to Prior Work: some prior works should be added and discussed.

Reproducibility: No

Additional Feedback:


Review 3

Summary and Contributions: This this paper, the author uses knowledge distillation on federated learning task with unlabeled data or generated data. This method outperforms traditional methods such averaging parameters in different model or averaging prediction score. The author verified their method on different tasks such as CV and NLP.

Strengths: 1. The author propose a distillation framework for federated model fusion. The method outperforms the baseline FEDAVG. 2. The author show in extensive experiments on various CV/NLP datasets and models. 3. The author provided insights why their methods work better than baseline.

Weaknesses: 1. The baseline methods in related work part is too simple, only one or two sentences. It will be better for reader to follow if the author could compare their methods with baselines in detail. 2. Local training in the experiments uses a constant learning rate (no decay), no Nesterov momentum acceleration, and no weight decay. In this case the local model will be a sub-optimal model, especially for large dataset like Imagenet. Will the conclusion change if the lcoal model are better trained? 3.

Correctness: The method is clearly described and each part is not tricky. The methods do not relay on complicated parameter tuning, so it should be correct for me.

Clarity: This paper needs improve. In the experiment part, the author compared FedDF with FEDAVG, FEDPROX and FEDMA. However, the author has not compared their method with algos above in related work part. Also, seems FedDF have some similiar part with FedMD and Cronus, but not compared with these two algos in experiments.

Relation to Prior Work: Not really. After reading Section 2 and 3, I was still not clear which part in their method is novel, which part is common practice and which part is baseline.

Reproducibility: Yes

Additional Feedback:


Review 4

Summary and Contributions: The paper proposes a new algorithm FedDF to address several challenges of cross-device federated learning (FL). FedDF consolidates knowledge from client models via knowledge distillation on unlabeled or synthetic dataset on a central server. Compared to the existing FL algorithms, FedDF (1) reduces communication overhead which is a major bottleneck of FL, (2) improves handling of heterogeneity of client data and (3) applies ensemble learning which can handle heterogeneous client model architectures in FL efficiently without changing the local training procedure. This paper brings a methodological advance in FL both in terms of efficiency and performance without violating the privacy of client data, all of which are first-order concerns in FL. It has clear advantages compared to relevant works and shows an extensive set of results to support its claims on various settings.

Strengths: This work manifests solid understanding of key requirements and challenges of federated learning, and thus presents a practical solution with significant improvements. The contribution of this paper is formulating a robust, efficient training scheme in FL with extensive results and analysis, which is relevant to the NeurIPS community. They provide sufficient justifications about why the additional computations are negligible in practice and why the reduced number of communication rounds and the ability to handle architecture heterogeneity of FedDF matter more. The authors analyzed its contribution from various angles including efficiency, utilizing heterogeneous computation resources of clients, robustness on the choice of distillation dataset, and handling heterogeneous client data by mitigating quality loss of batch normalization with different data distributions. The results are sensible and believable. The authors provided experimental results in various settings to demonstrate the advantage of FedDF: they tried (1) different models with varying capacities (ResNet-8/20, binarized ResNet-8, VGG, MobileNetV2, ShufleNetV2, DistilBERT), (2) multiple datasets with different number of classes (CIFAR-10/100, ImageNet, STL-10, AG News, SST2), (3) multiple values of alpha for controlling non-i.i.d.-ness of client data, and (4) repeated experiments over 3 random seeds. The authors provide in-depth analysis for each of the two components of FedDF by (1) providing theoretical bounds of the generalization performance of ensemble and (2) studying the effect of the distillation dataset in terms of number of classes, number of distillation steps, size of distillation dataset, and whether random noise could achieve the same performance. The quantified advantages (e.g. the number of communication rounds and accuracy) as well as its unquantified advantages (e.g. not affecting the local training procedure and being compatible with other orthogonal works) of FedDF seem like a solid advance in federated learning.

Weaknesses: This paper utilizes much less total clients in each experiment compared to original FedAVG paper (20 vs. 100). I understand that it could be difficult to get the computational resources required, but it can be too limited compared to a realistic scenario where billions of devices might participate. There are multiple occasions where the paper seems to contradict its own claims. While the authors show that FedDF is robust to choice of distillation dataset, they criticize FedMD and Cronus could affect training in an undesirable way by involving other datasets without providing any proof. In addition, they claim that they can improve the robustness or defend against attacks by using Cronus but they already criticized it for affecting the local training and argued that FedDF is better. Moreover, they cited “Overcoming Forgetting in Federated Learning on Non-IID Data” as an orthogonal approach to improve handling non-i.i.d.-ness of client data in addition to FedDF but it involves a penalty term to encourage all models to converge to a shared optimum which hinders the diversity of local models which FedDF benefits from. If these points are not addressed then it undermines (1) its contribution compared to FedMD and Cronus and (2) compatibility of FedDF with other orthogonal relevant works. The search space of hyperparameters seems limited. The “initial search grid” of learning rate is only 4 values between 1.5 and 0.01, and it remains unclear what kind of criteria is used for “extending search grid” when “the best hyperparameter” does not “lie in the middle of our search grid”. In comparison, the original FedAVG paper tuned learning rate in the original paper on a “wide grid” since SGD is sensitive to learning rate, and the original FedMD paper used 1e-03 with adam on CIFAR-100. Higher learning rate might not always reach better performance faster if it keeps missing good minima for many epochs. Limited search space of learning rates in this paper could have led to suboptimal performance of the models in terms of speed and accuracy. I am aware that the authors tried LR decay for FedAVG in the supplement, but the exact range and algorithm used remains unclear. Adam optimizer with the learning rate of 2e-3 is used for ensemble distillation from the server side, which uses not only much lower learning rate than the search space of local client training but also involves adaptive learning rate. Therefore it remains unclear if the benefit of FedDF is coming from the algorithm itself or being able to utilize a better learning rate in the process of training the model. For table 2 and 4, the authors did not provide performance of FedDF with group normalization which would have shown approximately how much performance gain is due to mitigating the quality loss of batch normalization. If this result is missing then the source of performance gain remains ambiguous (it could have been due to the low learning rate of local distillation, for example). -- Response to rebuttal: I no longer see any contradiction regarding the related works. (1) The clarification about their footnote 5 successfully clarified that it was the techniques mentioned in cronus paper, but not the cronus itself, was claimed applicable to FedDF. (2) Regarding the question about how “Overcoming Forgetting in Federated Learning on Non-IID Data” can be compatible with FedDF, the authors showed compatibility with FedProx instead. This is not an exact answer to the original question, but close enough because both “Overcoming Forgetting” paper and FedProx use an additional term to reduce diversity of local models which FedDF benefits from. They show that FedDF with prox reaches better performance than FedDF without prox, although marginally. The authors provided more experimental results to answer most of the questions I raised. In the rebuttal paragraph titled “[R5: Learning rate]”, they provide a justification why their LR grid was sufficient and provides additional results with lower LR and LR scheduling. They provided their new results in terms of accuracy rather than the number of communication rounds, which was the original concern. It is not clear if they are providing the best possible performance or the performance at the same number of communication rounds. Nonetheless, it demonstrates the advantage of FedDF compared to other methods either way. Performance of FedDF with group normalization is still not provided in the rebuttal. If the performance gain with GN is shown to be smaller in FedDF compared to other FL methods, then it would have provided additional support for the authors’ claim that FedDF with BN alleviates quality loss brought by BN.

Correctness: There is no obvious flaw in their design of FedDF. Requirements of FL such as preserving user privacy and minimizing communication overhead is satisfied. It doesn’t violate requirements of machine learning either (e.g. preventing information leak). Their method of simulating non-i.i.d.-ness of client data using Dirichlet distribution has been shown to be an effective method of investigating the effect of client data distribution in “Measuring the Effects of Non-Identical Data Distribution for Federated Visual Classification”. I did not detect any obvious difference between the algorithm proposed in the paper and the provided code.

Clarity: This paper is mostly coherent and explains what the readers need to know about FL before diving into details. They provide enough explanations for readers to understand the advantages FedDF provides in the context of FL. There are some parts that could have been more clear, though. When referring to figure 4 at the end of section 4, the authors say “gap between the fused model and the ensemble one widens when the training dataset contains a much larger number of classes” but I don’t think I observe this trend from figure 4. The authors could have pointed out specific numbers they used to generate this claim. In addition, the paper says in the introduction that “applying ensemble learning techniques directly in FL is infeasible in practice due to the large number of participating clients” and that “storing a different model per client on the server is not only impossible due to memory constraints, but also renders training and inference inefficient.” However, the paper introduces an approach which “distills the knowledge of all received client models” and “the fusion takes place on the server side”. This could be confusing at first: the readers might think that, if all received client models are stored in the server for FedDF, then it should still suffer from the memory constraints and inefficiencies discussed earlier. I think what the authors mean is that naive ensemble learning techniques will require saving model weights of all clients, including those not participating in the current round. But this was not immediately clear.

Relation to Prior Work: What’s novel about this paper is not the concept of applying knowledge distillation to FL or distributed training, as shown in works cited (e.g. FD). Rather, the contribution of this paper is formulating a robust, efficient training scheme with extensive results and analysis which is significant enough. The authors successfully discuss the advantages of their proposed algorithm compared to prior works in that regard. In comparison to FedAVG, FedProx and FedAVGM, FedDF has advantages of efficiency and being able to handle heterogeneity of client data and models better as shown in their results under various settings. Compared to FedMD and Cronus, FedDF has an advantage of leaving the local training unaffected. In addition, they point out that FedDF is compatible with some prior works and they can be additionally applied to improve the robustness of FL. For example, differential privacy (“Differentially private federated learning: A client level perspective.”) could be applied in order to further ensure privacy of local client data.

Reproducibility: Yes

Additional Feedback: I understand the benefit of compiling main and supplement documents as one pdf file for easily referencing figures and tables from each other (as it’s done in the supplement submission, which contains both main paper and supplement in supplement pdf file). However, the main paper at the beginning of the supplement pdf file had a permuted ordering of reference numbers compared to the main pdf file (probably due to additional citations in appendix), which was confusing. It might reduce confusion to use consistent ordering of references between the two pdf files (e.g. by ordering references according to its appearance in the text rather than alphabetical order of author last names).

[Author Response · NeurIPS 2020]

We thank all reviewers for their time and their valuable feedback. We will add corrections/clarifications as suggested.
We would like to emphasize our contribution, as summarized by R5's thorough review: "What's novel about this paper
is not the concept of applying knowledge distillation to FL or distributed training. Rather, the contribution of this paper
is formulating a robust, efficient training scheme with extensive results and analysis which is significant enough."
**[R1: Misunderstanding on algorithm]** Our proposed FedDF is not the mentioned engineering solution. Each model
architecture groups acquires knowledge from logits averaged over *all* received models (line 14 in Alg. 3, Fig. 7) for next
FL round; thus mutual beneficial information can be shared *across architectures*. The $S_t$ at Line 13 in Alg. 3 is correct.
**[R1: No privacy concern in using GAN]** GAN training is not involved in all stages of FL and cannot steal clients'
data. Data generation is done by the (frozen) generator *before the FL training* by performing inference on random noise.
**[R1: Clarity of line 32]** "[A]pplying ensemble learning techniques directly..." refers to keeping weights of all received
models on the server and performing naive ensembling (logits averaging) for inference (line 30-31). In contrast, we
distill the knowledge of *all* received models to the server model and then drop all received models' weights.
**[R2: Quality loss and data heterogeneity]** Final performance for all methods with different non-i.i.d. de-
grees/models/datasets is shown in Tab. 2 & 3, and FedDF is consistently the best performing method. As mentioned
in the caption of Tab. 1, the fine-tuned test accuracy of centralized training (on all local data) is 86%. The data
heterogeneity issue in FL results in quality loss (e.g. Fig. 2), and thus 80% and 75% are reasonable targets.
Due to computational infeasibility, ResNet-8 with fine-tuned hyper-parameters is used to provide in-depth empirical
understanding. Better performance can be achieved by larger model capacity, but is orthogonal to the provided insights.
**[R2 & R4 & R5: Comments on preprints FedMD and Cronus]** We comment on the two closest approaches (FedMD
and Cronus), in order to address 1) Distinctions between FedDF and prior work (R4), 2) Privacy/Communication traffic
concerns (R2), 3) Omitted experiments on FedMD and Cronus (R2, R4, R5).
• Distinctions between FedDF and prior work. As discussed in the related work, most SOTA FL methods directly
manipulate received model parameters (e.g. FedAvg/FedAvgM/FedMA). To our best knowledge, FedMD and Cronus
are the only two that utilize logits information (of neural nets) for FL. The distinctions from them are made below.
• Different objectives and evaluation metrics. Cronus is designed for robust FL under poisoning attack, whereas
FedMD is for personalized FL. In contrast, FedDF is intended for on-server model aggregation (evaluation on the
aggregated model), whereas neither FedMD nor Cronus aggregates the model on the server.
• Different Operations. 1) FedDF, like FedAvg, *only* exchanges models between the server and clients (line 114),
without transmitting input data. In contrast, FedMD and Cornus rely on exchanging public data logits. As FedAvg,
FedDF can include privacy/security extensions and has the same communication cost per round. 2) FedDF performs
ensemble distillation with unlabeled data *on the server*. In contrast, FedMD/Cronus use averaged logits received
from the server for *local client training*.
• Omitted experiments with FedMD/Cronus. 1) FedMD requires to locally pre-train on the *labeled* public data, thus
the model classifier necessitates an output dimension of # of public classes *plus* # of private classes (c.f. the output
dimension of # of private classes in other FL methods). We cannot compare FedMD with FedDF with the same
architecture (classifier) to ensure fairness. 2) Cronus is shown to be consistently worse than FedAvg in normal FL (i.e.
no attack case) in their Tab. IV & VI. 3) We thoroughly evaluated SOTA baselines with the same objective/metric.
**[R4: Local training technique]** Our experimental setup is widely adopted in many other published FL papers. We
observe that techniques like learning rate decay and local momentum are orthogonal to the model aggregation. Fig. 12
showcases the ineffectiveness of learning rate decay during local training. Including local Nesterov momentum only
marginally improves all methods without affecting the conclusion; we will include omitted results for local momentum.
**[R5: Clarification on the "contradiction claims"]** FedMD and Cronus have no evaluations on their choices of training
data construction, thus it remains unclear (line 89-90) how local training gets affected. Some general robust approaches
reviewed in Cronus (e.g. Krum, Bulyan) can be adapted to exclude faulty client models for FedDF. These techniques
alone do not interfere with the local training. We include extra results to justify the compatibility of FedDF with
orthogonal work; *fine-tuned* proximal penalty (from FedProx) is used *locally* as suggested, on CIFAR-10 with ResNet-8
(setups in Fig. 2). For non-iid degree $\alpha = 1$, the results of FedDF v.s. FedAvg over three seeds are: w/ prox 80.56 v.s.
76.11 and w/o prox 80.27 v.s. 72.73; for $\alpha = 0.1$, we have w/ prox 71.64 v.s. 62.53, and w/o prox 71.52 v.s. 62.44.
**[R5: Learning rate]** Our learning rate tuning is actually sufficient, as our used STOA networks are much less sensitive
to learning rate, different from the classical CNN (w/o BN and w/o residual connection) used in the original papers of
FedAvg and FedMD. The initial grid $\{1.5, 1, 0.5, 0.1, 0.05, 0.01\}$ loosely covers good SGD learning rates and can be
extended to scales such as $\{0.005, 0.001\}$ whenever necessary. A more fine-grained tuning only marginally improved
the results of all methods and did not affect our conclusions. The learning rate decay used in appendix (i.e. decay by 10
at 50% and 75% of the local training epochs) follows the general scheme as in many published papers.
To distinguish the benefits of FedDF from the small learning rate or Adam optimizer, we report the results of using
Adam (2e-3) for both local training and model fusion (over three seeds), on CIFAR-10 with ResNet-8 (setups in Fig. 2).
For $\alpha = 1$, we have 80.27 v.s. 72.73 (local training via SGD) and 83.32 v.s. 78.13 (local training via Adam); for $\alpha = 0.1$,
the results of FedDF v.s. FedAvg are 71.52 v.s. 62.44 and 72.58 v.s. 62.53 respectively. Improving the local training
through Adam might help FL but the benefit vanishes with higher data heterogeneity (e.g. $\alpha = 0.1$). Performance gain
from FedDF is robust to data heterogeneity and also orthogonal to effects of learning rates and Adam. As a side note,
FedDF uses the common learning rate magnitude for Adam (different from SGD's 1e-1) and is not small.

[Meta-Review · NeurIPS 2020]

I recommend this paper for acceptance. The paper is on an important and a timely topic and is above the quality bar necessary for acceptance. Although the reviewers had some concerns, the rebuttal clarified their most burning questions. I also thought that the more critical reviews were the less informed ones. Having said that, I strongly suggest to take all comments of the reviewers into account to improve the quality of the camera-ready version, mostly with respect to the organization, the clarity of the paper (including the description of the related work) and including the results provided in the rebuttal.